# New Insights into Neutrophil Extracellular Trap (NETs) Formation from Porcine Neutrophils in Response to Bacterial Infections

**DOI:** 10.3390/ijms23168953

**Published:** 2022-08-11

**Authors:** Marta C. Bonilla, Oriana N. Quiros, Michael Wendt, Isabel Hennig-Pauka, Matthias Mörgelin, Maren von Köckritz-Blickwede, Nicole de Buhr

**Affiliations:** 1Department of Biochemistry, University of Veterinary Medicine Hannover, Foundation, 30559 Hannover, Germany; 2Research Center for Emerging Infections and Zoonoses (RIZ), University of Veterinary Medicine Hannover, Foundation, 30559 Hannover, Germany; 3Escuela de Medicina Veterinaria, Universidad Nacional, Heredia 40104, Costa Rica; 4Clinic for Swine, Small Ruminants and Forensic Medicine and Ambulatory Service, University of Veterinary Medicine Hannover, Foundation, 30173 Hannover, Germany; 5Field Station for Epidemiology, University of Veterinary Medicine Hannover, 49456 Bakum, Germany; 6Colzyx AB, SE-223 63 Lund, Sweden

**Keywords:** neutrophils, neutrophil extracellular traps (NETs), vesicular NETs, pigs, *Actinobacillus pleuropneumoniae* (*A.pp*), *Streptococcus* (*S*.) *suis*

## Abstract

*Actinobacillus pleuropneumoniae* (*A.pp*, Gram negative) and *Streptococcus* (*S*.) *suis* (Gram positive) can cause severe diseases in pigs. During infection, neutrophils infiltrate to counteract these pathogens with phagocytosis and/or neutrophil extracellular traps (NETs). NETs consist of a DNA-backbone spiked with antimicrobial components. The NET formation mechanisms in porcine neutrophils as a response to both of the pathogens are not entirely clear. The aim of this study was to investigate whether *A.pp* (serotype 2, C3656/0271/11) and *S. suis* (serotype 2, strain 10) induce NETs by NADPH oxidase- or CD18-dependent mechanisms and to characterize phenotypes of NETs in porcine neutrophils. Therefore, we investigated NET induction in porcine neutrophils in the presence and absence of NET inhibitors and quantified NETs after 3 h. Furthermore, NETosis and phagocytosis were investigated by transmission electron microscopy after 30 min to characterize different phenotypes. *A.pp* and *S. suis* induce NETs that are mainly ROS-dependent. *A.pp* induces NETs that are partially CD18-dependent. Thirty minutes after infection, both of the pathogens induced a vesicular NET formation with only slight differences. Interestingly, some neutrophils showed only NET-marker positive phagolysosomes, but no NET-marker positive vesicles. Other neutrophils showed vesicular NETs and only NET-marker negative phagolysosomes. In conclusion, both of the pathogens induce ROS-dependent NETs. Vesicular NETosis and phagocytosis occur in parallel in porcine neutrophils in response to *S. suis* serotype 2 and *A.pp* serotype 2.

## 1. Introduction

Neutrophils are one of the most important cells in the immune defense against pathogens [1,2]. These cells can counteract pathogens with different antimicrobial mechanisms, as, for example, phagocytosis, degranulation or the release of neutrophil extracellular traps (NETs) [3,4]. NETs are extracellular structures formed by nuclear DNA from neutrophils, antimicrobial peptides, and enzymes [5,6]. There are two ways in which it is described that neutrophils can release NETs with nuclear DNA: 1. The “suicidal” NETosis; and 2. The “vital” NETosis. During “suicidal” NETosis, the activated neutrophil converts oxide to superoxide by NADPH oxidase (nicotinamide adenine dinucleotide phosphate oxidase). After receiving an activation signal, the different subunits of the NADPH oxidase (NOX2) from the membrane and the cytoplasm assemble into an active oxidase, which finally leads to the production of reactive oxygen species (ROS) [7,8]. Furthermore, the ROS activates peptidyl arginine deiminase 4 (PAD4), and is therefore involved in the “suicidal” NET release [9]. Cytokines, such as IL-8, pathogens or reagents, as, for example, phorbol 12-myristate 13-acetate (PMA), can induce NETs by activating the ROS production [10,11]. The production of ROS is initiated leading to the de-condensation of the chromatin and to the disintegration of the granules and nuclear membranes. Afterwards, the granular proteins, such as myeloperoxidase or elastase, interact and adhere to the DNA [5,12]. Finally, the cell membrane disrupts and the components are released into the extracellular space. This mechanism is described as an antimicrobial cell death, as the neutrophil dies, releases antimicrobial substances but cannot perform other defense mechanisms, such as phagocytosis or transmigration [12,13]. The “vital” NETosis is described as a vesicular mechanism. Here, the DNA from the nucleus and antimicrobial components from the granules are mixed in the small vesicles that migrate to the cell membrane, where they are finally released [14,15,16]. The neutrophil cell membrane stays intact during the whole process, and therefore the neutrophil is still alive and can undergo phagocytosis or can transmigrate [17]. Although most of the reports indicate that the vesicular NET release occurs mainly in a ROS-independent manner [13,18,19], there is also a report of neutrophils releasing mitochondrial DNA (mtDNA) through the vesicles, and, by applying diphenyleneiodonium chloride (DPI), a complete block of mitochondrial DNA release was observed, suggesting that a ROS-dependent vesicular NETs’ release can also occur [20].

Some of the pathogens can be entrapped in NETs and are extracellularly killed by enzymes [21,22]. However, several pathogens are able to escape out of this mechanism by, for example, producing DNases, such as *Streptococcus* (*S*.) *suis* [23,24], or taking advantage of degraded NETs, such as *Actinobacillus pleuropneumoniae* (*A.pp*) [25].

The Gram-negative bacterium *A.pp* leads mainly to pneumonia [26] and the Gram-positive bacterium *S. suis* leads to meningitis, arthritis, and pneumonia [27,28,29] in pigs. Therefore, both of the pathogens can cause severe diseases in pigs, with the consequence of high economic losses [30,31]. Furthermore, both of the bacteria induce NETs in pigs [24,25,32], but the detailed mechanism is still unknown.

For a better understanding of the host–pathogen interaction, knowledge is needed on how the bacteria interact with the neutrophils and induce NETs. Neutrophils have receptors on the outer membrane that can be activated directly via bacteria, indirectly by different bacterial components, or by released cytokines during infection. Several of the receptors that are important in NET formation have already been identified, as, for example, TLR-4, TLR-2, CD11a, or CR3 [15,33].

Interestingly, NET formation was described by the binding of a bacterial toxin to CD18 on bovine neutrophils. CD18 is the β subunit from the β2 integrin, which is only found on leukocytes [34]. This β subunit (CD18) is constant and can pair with different α subunits (CD11a, CD11b, CD11c, or CD11d) [35,36]. After the infection of the bovine neutrophils with *Mannheimia haemolytica* (bovine respiratory tract bacteria), they release NETs, because the leukotoxin (LKT) [37] of *Mannheimia haemolytica* binds to CD18.

*A.pp* produces pore-forming exotoxins (Apx toxins) as one of the most important virulence factors, which, in the same way as LKT, belong to the family Repeats in ToXin (RTX) [38]. In *A.pp*, these exotoxins are known to bind to CD18 receptors [39]. Furthermore, they can lyse the respiratory tissue cells and cells circulating in the blood stream, such as neutrophils [39,40].

*S. suis* produces a pore-forming cholesterol-dependent cytolysin (Suilysin) [41,42,43]. The different effects of this toxin on neutrophils are discussed [44,45]. However, it is not known whether *A.pp* or *S. suis* can induce NETs by binding their toxins to CD18.

The usage of NET inhibitors can help to gain knowledge about the mechanisms involved when the bacteria induce NETs [5,46,47]. DPI has been reported to block NADPH oxidase and, as a consequence, ROS production and NET release [7,48,49]. It is known that lipopolysaccharides (LPS), a cell wall component of Gram-negative bacteria [50], induce ROS-dependent NETs [49].

To investigate the NET formation mechanisms, the detection of free DNA is one of the most used NET markers [51]. However, the quantification of free DNA as a NET marker has some limitations; as, for example, it is not possible to differentiate between necrosis and NETosis. Furthermore, the frequently used PicoGreen assay to detect free DNA lacks the sensitivity to measure a relatively small amount of NETs [52]. Therefore, free DNA is not a specific NET marker, and an analysis based on visualization by NET-specific antibodies and immunofluorescence microscopy is recommended [11]. A more expensive, but also more detailed, NET analysis is possible, using scanning or transmission electron microscopy (SEM or TEM) [53]. Both of the techniques allow for the evaluation of the cellular and fine structures in a nanometer scale resolution [54]. TEM is used to observe the internal cell structures. Meanwhile, SEM is used to observe the cell surface [55].

The objectives of this study were to determine whether *A.pp* and *S. suis* induce NAPDH-dependent NETs and/or via CD18 activation. Furthermore, we investigated whether *A.pp* and *S. suis* induce “vital” NETosis in porcine neutrophils, and whether these neutrophils are still able to undergo phagocytosis.

## 2. Results


***A.pp* and *S. suis* induce NADPH oxidase-dependent NETs and *A.pp* also via CD18 activation**


To determine how *A.pp* and *S. suis* induce NETs, we conducted NET induction assays with porcine neutrophils in the presence of DPI or an antiCD18a antibody to block the NET release via CD18 activation. First, we conducted screening assays to adjust the experimental settings. Different infection doses of *A.pp* (multiplicity of infection (MOI) of 0.5, 1, and 2) induced NET formation after 3 h and no difference was observed (Figure A1, Appendix A). Therefore, a range of MOI = 0.5–2 for *A.pp* (freshly grown) was used in the following assays. Secondly, we investigated whether the addition of 2% fetal bovine serum (FBS) reduces the spontaneous NET release in porcine neutrophils, as described by Kamoshida et al. [56]. A reduction in the spontaneous NETs could help to identify small differences after treatment. Compared to a sample without FBS, we did not identify fewer NETs by qualitative analysis in the untreated neutrophils in the presence of FBS, but instead observed another critical phenomenon. The porcine neutrophils that were incubated with FBS showed a tendency to clump together (Figure A2, Appendix A). Therefore, all of the following assays were conducted in the absence of FBS. In a following screening experiment, the cell-free DNA as a NET marker [51] was analyzed in the supernatants of *A.pp* NET induction assays. We observed an increasing amount of cell-free DNA over time in the supernatant of the neutrophils that were treated with methyl-β-cyclodextrin (CD 1 h = 0.72 µg/mL and CD 3 h = 1.3 µg/mL), even in the presence of NET inhibitors (Figure A3A,B, Appendix A; 1 h _antiCD18_ = 0.69 µg/mL, 3 h _antiCD18_ = 1.28 µg/mL, and 1 h _DPI_ = 0.56 µg/mL, 3 h _DPI_ = 1.16 µg/mL). In the supernatant of the *A.pp*-treated neutrophils, an increasing amount of cell-free DNA was detected over time (1 h = 0.85 µg/mL and 3 h = 1.2 µg/mL), but, interestingly, already after 1 h incubation, less cell-free DNA was released by the neutrophils under the DPI treatment and the *A.pp* infection, compared to the samples without DPI (Figure A3A, Appendix A; 1 h _DPI_ = 0.6 µg/mL). This difference was significant after 3 h DPI treatment and *A.pp* infection (3 h _DPI_ = 0.74 µg/mL, *p* = 0.002). Furthermore, after 3 h, significantly less cell-free DNA was released by the *A.pp* infected neutrophils in the presence of antiCD18a compared to the neutrophils without inhibitor (Figure A3B, Appendix A; 3 h _antiCD18_ = 1.1 µg/mL; *p* = 0.008).

Based on the screening results, we conducted, in the following steps, the NET quantification analysis by immunofluorescence microscopy after 3 h infection with *A.pp* and *S. suis* (Figure 1). The neutrophils released significantly fewer NETs after *A.pp* and *S. suis* infection in the presence of DPI compared to the samples without an inhibitor (DPI: *A.pp*: *p* < 0.0001 and *S. suis*: *p* < 0.0001) (Figure 1B).

Furthermore, we investigated the role of CD18 activation in NET formation. The *A.pp* induced CD18-dependent NETs (Figure 1B, *p* = 0.0085). Additionally, a significant difference was observed between the antiCD18a and DPI pretreated and *A.pp* infected neutrophils (*p* = 0.0218) (Figure 1B).

Whereas the antiCD18a partially blocked the NET release by *A.pp*, another result was observed after the *S. suis* infection of the antiCD18a incubated neutrophils. Surprisingly, the pre-incubation with antiCD18a significantly increased the NET release by *S. suis*-infected neutrophils (*p* = 0.0283), indicating that the *S. suis* did not induce NETs via CD18. A significant difference was observed between the antiCD18a and DPI pretreated and *S. suis* infected neutrophils (*p* < 0.0001) (Figure 1B).

In summary, we identified that, after *A.pp* and *S. suis* infection, the porcine neutrophils released NETs NADPH oxidase dependently. Furthermore, the *A.pp* NET release by the porcine neutrophils was shown to be CD18-dependent.


***A.pp* and *S. suis* infected porcine neutrophils release vesicular NETs and undergo phagocytosis**


In the following step, we analyzed whether the neutrophils react in the absence of antibodies on Gram-negative and Gram-positive bacteria in comparable ways regarding their NET release and phagocytic activity. Therefore, we analyzed the neutrophils with TEM after 30 min infection for phagolysosomes, and NET formation by immune gold-labeling of bacteria (phagolysosomes in serial Section 1) and NE, H3cit (NETs in serial Section 2). In the overview pictures of the *A.pp-* and *S. suis*-infected neutrophils, the neutrophils with cytoplasmic NETs (NE and H3cit positive) were identified (Figure 2A and Figure 3A, area 1), as these neutrophils are filled with decondensed chromatin. It is conceivable that the content can be released into the extracellular space after a longer incubation time by breaking off the outer membrane, which is described as a sign for “suicidal” NETosis. Furthermore, the neutrophils with H3cit- and NE-positive vesicles were identified inside neutrophils by immune gold-labeling in serial Section 2 (Figure 2A and Figure 3A, area 2), and interestingly adhering at the outer membrane of the neutrophil (Figure 2A and Figure 3A, area 3). We could clearly identify the phagolysosomes containing *A.pp* or *S. suis* using the immune-gold labeling of bacteria in the serial Section 1. Interestingly, after *A.pp* and *S. suis* infection, we identified two different types of neutrophils in the same samples. The neutrophil “type separate = type S” showed phagolysosomes that were positive for only bacteria (serial Section 1) and separate vesicles that contained H3cit and NE (serial Section 2) (Figure 2B and Figure 3B). The neutrophil “type merged = type M” showed phagolysosomes that were positive for bacteria (serial Section 1) and contained H3cit and NE, but no separate H3cit/NE positive vesicles could be found inside such neutrophils (serial Section 2) (Figure 2C and Figure 3C). Furthermore, we identified occasional nuclear membrane blebbing positive for H3cit and NE (Figure 2D and Figure 3D), however the nuclear vesicles present in the cytoplasm, apparently already detached from the nucleus or released from the neutrophil, were more frequently identified in these samples.

Finally, we quantified whether *A.pp* and *S. suis* comparably induced “suicidal” and “vesicular” NETs after a short infection time compared to a negative control (Figure 4). The neutrophils that showed a loss of intact cytoplasmatic structure with a release of nuclear DNA were counted as neutrophils undergoing “suicidal” NETosis. Indeed, the *A.pp-* and *S. suis*-infected neutrophils released NETs in comparable amounts from the nuclei (Figure 4B). In addition, we identified a comparable amount of NET surface coverage in the infected cells (Figure 4C). Furthermore, *A.pp* and *S. suis* significantly induced higher numbers of vesicles (characteristic of viable/vesicular NETosis) per neutrophil compared to an unstimulated control. Interestingly, *S. suis* induced significantly more vesicles compared to *A.pp* (Figure 4D).

In summary, already, after only 30 min of infection, different NET/neutrophil types could be observed.

## 3. Discussion

The NET formation by neutrophils has been described for over 15 years [6]. Over the last few years, several new insights regarding NET release were identified, for example, the differentiation between “suicidal” and “vital” NETosis [12,13]. As there is still a lack of knowledge about the mechanisms of NET release, more research is needed to focus on investigating whether “vital” and “suicidal” NETosis occur in vivo [57]. Interestingly, the vesicular NET formation could not only be detected after bacterial infections, but it was also detected ex vivo by our research group in the neutrophils from the vitreous body fluid of a horse with an inflammatory eye disease (equine recurrent uveitis) [58]. Furthermore, the stimulation of neutrophils with the equine cathelicidin 1 (eCATH 1) induced vesicular NET formation. This underlines that the vesicular NET release is probably a frequent mechanism in infectious and non-infectious diseases. Therefore, a better understanding of this mechanism is urgently needed.

Pilsczek et al. showed that the “vital” NETosis induced by *Stapyhlococcus aureus* is ROS-independent [19]. Therefore, it is unclear how the fast vesicular NETosis is initiated by bacteria. We observed that *A.pp* and *S. suis* induce ROS-dependent NETs in porcine neutrophils. For *S. suis*, this is in accordance with a study that was conducted with murine neutrophils [32]. However, we additionally observed that DPI cannot block the whole NET release after *S. suis* and *A.pp* infection (Figure 1). This also indicated a ROS-independent NET formation. It is possible that other receptors, such as CD18, are activated to release NETs. In the case of *A.pp*, it is likely that the Apx toxins activate this receptor. Further studies are needed to identify the initiating factors for the vesicular NET formation. One assumption is that either bacteria or bacterial virulence factors bind to the receptors at the outer membrane and they are thereby activated. Another assumption is that after phagocytosis, the intracellular processes are initiated and start the formation of the vesicular NETs. Therefore, the neutrophils need to undergo phagocytosis and NET formation in parallel.

The first step would be to understand the NET formation in vitro and to establish methods to record “vital” and “suicidal” NETosis.

One critical point is that, until now, only TEM analysis was used to reliably identify the vesicular NETosis. The vesicular NETosis was previously described after human neutrophils were infected with *Staphylococcus aureus* [19]. In the study by Pilsczek et al., a nuclear membrane blebbing was observed with TEM images. This describes a separation of the inner nuclear membrane from the outer nuclear membrane. Inside the space between these two membranes, DNA with nucleosomes were observed, referred to as “beads on a string”. Indeed, we also occasionally observed a separation of the two nuclear membranes in our samples, after being infected with *S. suis* and *A.pp* for 30 min. However, the nuclear vesicles present in the cytoplasm, apparently already detached from the nucleus, occurred more frequently in our samples (Figure 2D and Figure 3D).

When comparing “suicidal” and “vital” NETosis in our investigated samples, we identified that both of the pathogens induced “suicidal” NETosis (NET release from the nuclei, Figure 4B) to comparable amounts after 30 min infection with an MOI of 10. As this was significantly different from an uninfected control, the bacteria induced this “suicidal” NETosis. Nevertheless, there are indications that the analysis by TEM is more sensitive compared to the analysis by immunofluorescence microscopy. The porcine neutrophils infected with *S. suis* (MOI = 2) were analyzed in one of our own studies by immunofluorescence microscopy, and after 30 min showed a significant NET induction. However, only around 12% of the neutrophils were counted as NET-releasing cells. This could be explained by either the lower MOI or by the fact that in immunofluorescence microscopy, not all of the NET positive cells can be clearly identified.

In addition to the “suicidal” NET formation, we observed the “vital” NETosis by TEM and could detect a significantly higher number of nuclear vesicles per neutrophil after *S. suis* infection (Figure 4D). One explanation could be that the different toxins of *A.pp* (Apx toxins) and *S. suis* (suilysin) act differently on neutrophils and thereby could influence the NETosis process of neutrophils. The *A.pp* strains that produce Apx toxins provoke an oxidative burst of porcine neutrophils after 25 min [59], and high concentrations of Apx toxins are toxic for neutrophils [60]. Suilysin, the toxin of *S. suis*, is described as cholesterol-dependent cytolysin that forms pores in the cells, including the neutrophils. After 240 min, only the suilysin-producing strains are cytotoxic to the neutrophils [61]. However, whether these toxic effects of Apx toxins and suilysin lead to the observed differences of vesicular NETs needs to be investigated in future studies, with the inclusion of the Apx toxin deletion mutants of *A.pp* and suilysin negative strains or deletion mutants.

Independent of the vesicles that were observed and after 30 min of infection with both of the bacteria, we identified phagolysosomes and separate vesicles that were H3cit and NE positive in the neutrophils (type “S” neutrophils) (Figure 2B and Figure 3B). This indicates how fast the process of phagocytosis and vesicular NETosis starts. It is described that the phagosomes may fuse with NE and myeloperoxidase from granules, but this would not explain our findings of H3cit in the phagolysosome (type “M” neutrophils). However, we have not yet been able to identify if phagocytosis and vesicular NETosis start in parallel, or if one initiates the other. Future studies should focus on answering the question of at what moment a neutrophil or a specific neutrophil population can undergo “vital” NETosis and/or phagocytosis or change the strategy and exclusively perform “suicidal” NETosis. One other possibility is that every neutrophil can undergo all of the mechanisms and it is only a question of at which time-point the neutrophils were observed. Therefore, an observation of individual neutrophils with live cell imaging techniques would be needed, that can visualize the vesicles and bacteria inside a neutrophil. As live cell imaging with TEM is not yet possible [62], the only option would be high resolution immunofluorescence microscopy. Nonetheless, there are several technical challenges that need to be solved: 1. The assays need to be performed with fluorescence-labeled bacteria and the influence of this labeling on phagocytosis and NETosis needs to be investigated first; 2. The vesicles we observed in this study have a mean size of only 200–300 nm; 3. A specific marker for the vesicular NETs (H3cit and NE positive vesicles) working in live-cell imaging needs to be found and established. The study by Thiam et al. characterized NETosis in human and murine neutrophils and differentiated HL-60 neutrophil-like cells (dHL-60) by analyzing the plasma membrane microvesicles with high-resolution time-lapse microscopy [63]. The authors observed that NETosis took place in a stepwise manner, and they showed that the yeast *Candida albicans* was actively phagocytosed by neutrophils, and that these neutrophils additionally completed NETosis quickly. It was described that the neutrophils can sense the size of a microbe and for this reason can undergo fast phagocytosis to encounter small pathogens, such as bacteria or slow “suicidal” NETosis, to fight against large pathogens, such as *Candida albicans* hyphae [64]. Furthermore, it was discussed that the neutrophils, after the phagocytosis of the apoptotic cells, keep calm and do not release NETs [65,66]. Nevertheless, our data show that neutrophils phagocytose *S. suis* and *A.pp* and afterwards, or in parallel, form and release the vesicular NETs (Figure 2 and Figure 3). Furthermore, we already observed a high number of neutrophils with “suicidal” NETosis after 30 min (Figure 4). Our data do not reveal whether these neutrophils phagocytosed already before the bacteria or whether the bacteria, that were extracellularly entrapped in the NETs, were phagocytosed together with the NETs’ structures. This could also explain the “type M” neutrophils. To clarify this, other techniques, as described above, are needed. Independent of this limitation of our study, we showed that the porcine neutrophils can react as multi-tasking cells to an infection with *A.pp* or *S. suis*. Nevertheless, the neutrophils from the same donor do not react equally to one stimulus and seem to have differences in the chronological sequence.

## 4. Materials and Methods

### 4.1. Collection of Porcine Blood

The collection of blood from healthy pigs was registered at the Lower Saxonian State Office for Consumer Protection and Food Safety (Niedersächsisches Landesamt für Verbraucherschutz und Lebensmittelsicherheit, No. 33.9-42502-05-18A302). It was conducted in line with the recommendations of the German Society for Laboratory Animal Science (Gesellschaft für Versuchstierkunde) and the German Veterinary Association for the Protection of Animals (Tierärztliche Vereinigung für Tierschutz e. V.) (http://www.gv-solas.de, accessed on 28 July 2022).

The donor pigs were kept at the Clinic for Swine, Small Ruminants and Forensic Medicine and Ambulatory Service or at the Research Center for Emerging Infections and Zoonoses (RIZ) of the University of Veterinary Medicine Hannover, Foundation, Hannover, Germany. A maximum of 30 mL of fresh heparinized blood was collected from healthy pigs in S-Monovette^®^ Lithium-Heparin 9 mL tubes (Sarstedt AG and Co. KG, Nümbrecht, Germany). The blood was immediately used to isolate the neutrophils.

### 4.2. Growth Conditions of A.pp and S. suis

In this study, *A.pp* serotype (ST) 2 C3656/0271/11 was used. This strain was isolated during routine diagnostics at the Institute of Microbiology, University of Veterinary Medicine Hannover, Foundation, from the lung tissue of a diseased fattening pig during an *A.pp* outbreak [67]. The isolated strain was stored in the strain collection at −80 °C. A streak out was made on boiled blood agar plates with nicotinamide adenine dinucleotide (NAD) and incubated for 24 h at 37 °C with 5% CO_2_. The streak out was stored at 4 °C and used for a maximum period of 7 days. The *A.pp* ST 2 was grown in medium, as described previously [25]. Briefly, the bacteria were grown in pleuropneumonia, similar to organism medium (PPLO) supplemented with 0.1% Tween 80 and Isovitale X to the late exponential growth phase. This growth phase is characterized by an optical density (OD_600nm_) of 0.60 ± 0.01. The bacteria were washed twice in 1× PBS (Lipopolysaccharides (LPS)-free). The fresh grown *A.pp* was used in the NET inhibition assay (Figure 1; Figure A3, Appendix A). The pellet was finally adjusted to an OD_600nm_ = 0.6 with 1× PBS LPS free and used fresh. In all of the other presented data, the bacteria were grown to late exponential growth phase OD_600nm_ = 0.60 ± 0.01, washed with 1× PBS and frozen in liquid nitrogen and stored at −80 °C until usage (production of working cryostocks). A thawed working cryostock was used only once.

*Streptococcus* (*S.*) *suis* cps type 2 strain 10, kindly provided by Hilde Smith (Wageningen, GE, the Netherlands) [68], was stored in the strain collection at −80 °C. A streak out was made on Columbia agar with 5% sheep blood (Merck GmbH, Darmstadt, Germany) and incubated for 24 h at 37 °C. The streak out was stored at 4 °C and used for a maximum period of 7 days.

For preparing the working cryostocks, the bacteria were freshly grown in tryptic soy broth (TSB) without dextrose (Becton Dickinson, 286220, Franklin Lakes, NJ, USA) until the early stationary growth phase, as described before [69]. The culture was mixed with glycerol (final concentration of 15%) and the aliquots were immediately frozen in liquid nitrogen and stored at −80 °C until usage. Each working cryostock was used only once.

### 4.3. Neutrophil Isolation

The porcine neutrophils were purified from heparinized blood, as previously described [25,51]. A density gradient with Biocoll (1.077 g/mL, L6115; Biochrom, Berlin, Germany) or Biocoll^®^ (1.077 g/mL; Bio&SELL GmbH, BSL6115, Nürnberg, Germany) was conducted. The isolated neutrophils were resuspended in 1 mL cold Roswell Park Memorial Institute (RPMI) 1640 Medium, without phenol red (11835063 Gibco^TM^; Thermo Fisher Inc., Waltham, MA, USA). The cell number was adjusted with RPMI to the cell number depending on the experiment.

### 4.4. NET Inhibition Assay in Presence of A.pp and S. suis

For the NET inhibition assay, the cover slides (8 mm; Thermo Fisher Scientific, Waltham, MA, USA) were placed in 48 well plates (677102; Greiner Bio-One, Kremsmünster, Austria) and coated with poly-L-lysine (0.01% solution P4707; Sigma-Aldrich GmbH, Munich, Germany), following the manufacturer’s instructions, and handled afterwards as previously described [25]. In each well, 2 × 10^5^ neutrophils/100 µL were seeded. As a negative control, RPMI 1640 medium without phenol red was added. As positive controls, either 100 µL methyl-β-cyclodextrin (CD) (10 mM final concentration, C4555; Sigma-Aldrich GmbH) or 100 µL phorbol 12-myristate 13-acetate (PMA) (25 nM final concentration, 524,400; Sigma-Aldrich GmbH) was used. The neutrophils were infected with freshly grown *A.pp* ST 2 or *S. suis* cps type 2 strain 10 from cryostocks. Twice in 1 × PBS (Lipopolysaccharide (LPS) free), the washed *A.pp* were diluted 1:10 with RPMI. Due to variations in growth in the freshly grown *A.pp*, an MOI in a range of 0.5–2 was used in the NET inhibition assay. The *S. suis* were diluted in RPMI and an MOI of 2 was used.

In the NET inhibition assay, two inhibitors were separately used: 1. A mouse anti-pig CD18a (MCA1972GA; Bio-Rad Laboratories Inc., Hercules, CA, USA) with a final concentration of 1 µg/mL; 2. Diphenyleneiodonium chloride (DPI) (D2926-10MG; Sigma-Aldrich GmbH) with a final concentration of 10 µg/mL.

The neutrophils were seeded on the coated cover slides and incubated for 30 min at 37 °C 5% CO_2_ in the presence of the two inhibitors. Afterwards, the NET inducers were added (CD, PMA, *A.pp*, or *S. suis*) and the slides were centrifuged (370× *g*, 5 min) and incubated for 3 h at 37 °C 5% CO_2_. The final volume per well was 200 µL.

In one test experiment, the neutrophils from the same isolation were incubated with *A.pp* MOI = 0.5, 1, and 2 (from working cryostocks) for 3 h at 37 °C, 5% CO_2_ (Figure A1, Appendix A).

In another test experiment, the neutrophils from the same isolation were incubated with RPMI (negative control) in the presence or absence of 2% heat inactivated FBS (FBS SUPERIOR stabil^®^S0615; Bio&SELL GmbH, Feucht, Germany) for 3 h at 37 °C, 5% CO_2_ (Figure A2, Appendix A).

After incubation, the samples were fixed with paraformaldehyde (4% final concentration) and the plates were wrapped with parafilm and stored at 4 °C until the staining for the immunofluorescence microscopy analysis was conducted.

### 4.5. NET Staining

The NETs were stained as previously described [70]. After the permeabilization and blocking, the samples were incubated with the primary antibodies. The samples were stained with mouse monoclonal antibody (IgG2a) against DNA/histone 1 (MAB3864; 0.55 mg/mL diluted 1:1000 (*A.pp* NET inhibition) and 2.2 mg/mL diluted 1:4000 (*S. suis* NET inhibition and data presented in Appendix A); Millipore, Billerica, MA, USA). The *S. suis* NET inhibition samples were additionally stained with a polyclonal rabbit anti-human myeloperoxidase antibody (A039829-2; 3.3 mg, 1:309; Agilent, Santa Clara, CA, USA). All of the primary antibodies were incubated for 1 h at room temperature. For the isotype controls, a murine IgG2a (from murine myeloma, M5409-0.2 mg/mL, 1:364 Sigma-Aldrich GmbH) for both of the NET inhibition assays were used. A rabbit IgG (from rabbit serum, I5006, 1.16 mg, 1:108.75; Sigma-Aldrich GmbH) was additionally added for the *S. suis* NET assay. As the secondary antibody, a goat anti-mouse DyLight 488 (1:1000; Invitrogen, Carlsbad, CA, USA) (*A.pp* NET inhibition) and a goat anti-mouse Alexa 488Plus (1:500; A32723, 2mg/mL, Invitrogen) (*S. suis* NET inhibition) were used. In addition, the *S. suis* NET inhibition samples and NET induction assays presented in Figure 1 and Appendix A were stained with a goat anti-rabbit Alexa 633 (A21070, 2 mg, 1:500; Thermo Scientific, Waltham, MA, USA). The secondary antibodies were incubated for 1 h at room temperature in the dark. Afterwards, the samples were washed three times with 1× PBS (phosphate buffered saline) and once with distilled water. The samples were stained for 10 min (in the dark at room temperature) with a Hoechst 33,342 solution (1:1000, stock 50 mg/mL; Sigma-Aldrich GmbH). The slides were then washed three times with distilled water and embedded in 3–5 μL ProlongGold (without DAPI; Invitrogen). The samples were dried overnight at 4 °C and cover slips were surrounded with clear nail polish and stored at 4 °C in the dark until analysis.

### 4.6. Staining of Bound Anti CD18a on Neutrophils

The control samples from the inhibition assay were stained to confirm by immunofluorescence microscopy the binding of the CD18a antibody onto the surface of the neutrophils. The permeabilization and the blocking for the staining were performed as above. Myeloperoxidase was detected using a polyclonal rabbit anti-human myeloperoxidase (A039829-2, 3.3 mg, 1:309; Agilent, Santa Clara, CA, USA). For the isotype controls, a rabbit IgG (from rabbit serum, I5006, 1.16 mg, 1:108.75; Sigma-Aldrich GmbH,) and a IgG1 Isotype Control from murine myeloma (M5284, 0.2 mg, 1:200; Sigma-Aldrich GmbH) were used. The first antibodies were incubated for 1 h at room temperature. As secondary antibodies, a goat anti-mouse IgG (H + L) DyLight 633 (#35512, 1:500; Invitrogen) and a goat anti-rabbit IgG (H + L) cross-adsorbed Alexa Fluor 488 (A11008, 1:500; Invitrogen) were used. The secondary antibodies were incubated for 1 h in the dark and at room temperature. The processing of the samples was carried out as explained above. The images were taken using a Leica TCS SP5 AOBS confocal inverted-base fluorescence microscope with an HCX PL APO 40× 0.75–1.25 oil immersion objective (see Figure A5, Appendix A).

### 4.7. NET Quantification in Immunofluorescence Images

The NETs were quantified as previously described [71]. Six randomly chosen pictures were taken, using a Leica TCS SP5 AOBS confocal inverted-base fluorescence microscope with an HCX PL APO 40 × 0.75–1.25 oil immersion objective. The pictures were taken at predefined positions on two slides (see Figure A4, Appendix A). Only if an artefact (e.g., air bubble) was present was another adjacent area selected. The focus was set on the nuclei (blue channel). The cells were counted manually, using ImageJ software (version 1.52q; National Institute of Health, Bethesda, MD, USA) with the Cell counter plugin. All of the NET negative and positive neutrophils were counted. The neutrophils on the border of an image (not fully visible) were not counted at all. In each sample, a minimum of 300 cells were counted. A neutrophil was counted as positive if an evident off-shoot of DNA was visible, or if at least two of the following criteria were found: enlarged nucleus; decondensed nucleus; or blurry rim. The percentage of the NET-positive neutrophils was calculated (NET-activated cells). An average from the counted NET-activated cells from the six pictures from each sample was calculated.

### 4.8. Electron Microscopy of Neutrophils Infected with A.pp and S. suis

The neutrophils (1 × 10^6^/100 µL) were incubated in a 1.5 mL tube with RPMI 1640 Medium or *A.pp* (MOI = 10) or *S. suis* (MOI = 10) for 30 min at 37 °C, 5% CO_2_. The final volume per tube was 200 µL. After incubation, the samples were fixed with paraformaldehyde (4% final concentration) and stored at 4 °C. The samples were rebuffered with 2.5% (vol/vol) glutaraldehyde in 0.1 M sodium cacodylate (pH 7.2). They were then post fixed with 1% osmium tetroxide (wt./vol) and 0.15 M sodium cacodylate (pH 7.2) for 1 h at 4 °C, washed, and further processed for electron microscopy.

For the transmission electron microscopy, the fixed and washed samples were subsequently dehydrated in ethanol and further processed for standard Epon embedding, as described previously [58]. The ultrathin sections were stained with uranyl acetate (Laurylab, Saint Fons, France) and lead citrate (Laurylab). The sections were cut in 70 nm slices with an LKB ultratome and mounted on Formvar-coated copper grids. The immunolabeling of thin sections after antigen unmasking with sodium metaperiodate (Merck) [72] was performed as described previously [73], with the modification that Aurion-BSA (Aurion, Wageningen, the Netherlands) was used as a blocking agent. The following antibodies were used: gold-labeled Anti-Histone H3 (citrulline R2 + R8 + R17) antibody (1:80 diluted; H3cit, 5 nm gold; ab5103; Abcam, Berlin, Germany); anti-neutrophil elastase (1:80 diluted; NE, 10 nm gold; ab131260; Abcam, Berlin, Germany); gold-labeled polyclonal rabbit anti-*A.pp* St7 antibody [25] (1:50; *A.pp* 10 nm gold); and gold-labeled rabbit IgG anti-*S. suis* (1:500; *S. suis* 10 nm gold) [74].

In some of the experiments, 70 nm-thick serial sections were assessed for immunolabeling of the same cell compartments against different targets (Section 1: NET vesicles with H3cit and NE and Section 2: bacteria (*A.pp* or *S. suis*).

The images were recorded using a Philips/FEI CM100 BioTwin transmission electron microscope operated at a 60-kV accelerating voltage, with a Gatan Multiscan 791 charge-coupled device camera.

### 4.9. NET Quantification in EM Images

The NETs were quantified as follows: in total, the cellular profiles from 30 randomly selected fields on the thin sections were analyzed per sample for three different analysis points. For each analysis point, 30 different fields were assessed, as follows:NET release from the nuclei: loss of intact cytoplasmatic structure with release of nuclear DNA was counted as positive. The percentage of positive cells was calculated in relation to all of the observed cells investigated;NET surface coverage: the area covered with NET structures that was found in the extracellular environment of neutrophils was measured. The percentage indicated the fraction of the area covered by NETs in relation to the total area of the respective field;Nuclear vesicles per neutrophil: the number of the intracellular nuclear vesicles (H3cit and NE positive) were counted per neutrophil in the randomly selected fields. The analyzed neutrophils show an intact cytoplasmic structure with nucleus, double nuclear membrane, and intact granules.

### 4.10. Measurement of Cell-Free DNA in Supernatants of A.pp NET Assays

The assay was conducted as described above, without cover slides inside the wells. After adding *A.pp*, the samples were incubated for 1 h and 3 h. At the end of the incubation, the samples were treated with micrococcal nuclease (MN) (with a final concentration of 100 mU/200 µL, micrococcal nuclease from Staphylococcus aureus, N5386-200UN; Sigma-Aldrich GmbH) for 10 min at 37 °C. The samples were centrifuged (370× *g*, 5 min) and 150 µL of the supernatant was collected and frozen at −20 °C until further analysis.

The supernatants were analyzed to determine the amount of cell-free DNA in the samples. Therefore, a Quant-iT ™ PicoGreen ™ assay (Thermo Fisher; P11496 Invitrogen) was used as described previously [51]. The amount of DNA in each sample was calculated based on the standard curve.

### 4.11. Statistical Analysis

The data were analyzed using Excel 2019 (Microsoft) and GraphPad Prism 9.1.0 (221) and 9.2.0 (283) (GraphPad Software, San Diego, CA, USA). The data were analyzed for normal distribution using the Shapiro–Wilk test. The data were analyzed with the one-tailed paired Student’s *t*-test (free DNA) or Mann–Whitney test (immunofluorescence microscopy analysis), and one-way ANOVA followed by Tukey’s multiple comparison test (electron microscopy). The data are presented with mean ± SD and the differences between the groups were analyzed, as described in the figures’ legends (* *p* < 0.05, ** *p* < 0.01, *** *p* < 0.001, **** *p* < 0.0001).

## 5. Conclusions

The interaction of porcine neutrophils and Gram-positive (*S. suis*) and Gram-negative (*A.pp*) bacteria is an important host–pathogen interaction in the early phase of an infection. We observed that the porcine neutrophils, already after 30 min of infection with *S. suis* serotype 2 strain 10 and *A.pp* serotype 2, counteract the infection by phagocytosis, and in parallel by vesicular NETosis (“vital” NETs). Furthermore, other neutrophils already performed “suicidal” NETosis at the same time, which is characterized by NET release from the nuclei and NET structures that cover the surrounding surface of the neutrophils. In general, our data show that porcine neutrophils from one donor react to porcine pathogens in a short time, with more than one defense mechanism. Interestingly, some differences in the number of vesicles per neutrophil were identified, depending on the infecting bacteria. This indicates that the neutrophils do not react entirely in the same way with an “all in” reaction after bacterial infection, and that a bacteria-specific reaction of the neutrophils is triggered.

## Figures and Tables

**Figure 1 ijms-23-08953-f001:**
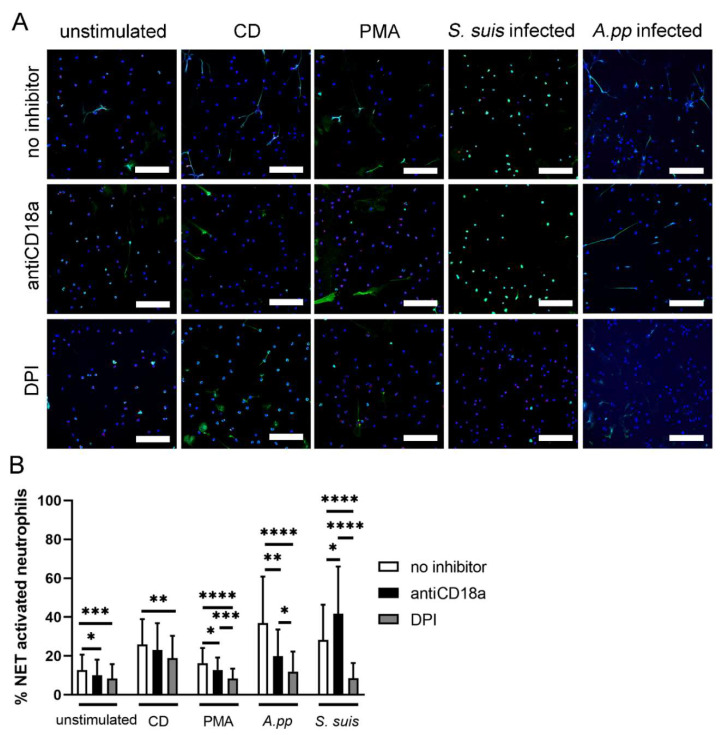
(**A**) Representative immunofluorescence images (overlay) of NET inhibition assays used to quantify activated cells are presented. In each experiment and for each sample, six randomly taken pictures from two individual slides were analyzed for quantification. All cells on the six pictures were counted and the mean of activated cells per stimulus was calculated and used for statistics. RPMI was used as unstimulated control. Methyl-β-cyclodextrin (CD) and phorbol 12-myristate-13 acetate (PMA) diluted in RPMI were used as positive controls. DPI and CD18a antibodies were used as inhibitors of ROS-dependent and independent NET release, respectively. Staining: Blue = DNA; green = DNA/histone-1-complexes; red = myeloperoxidase (except *A.pp* treated samples); scale bar = 100 µm. All settings were adjusted to respective isotype controls; (**B**) Statistical analysis of NET inhibition assay (3 h stimulation) sorted by stimulus. NET release by *A.pp* is significantly blocked by antiCD18a and DPI. NET release by *S. suis* is significantly blocked by DPI and significantly increased by antiCD18a co-incubated with *S. suis*. The data are presented with mean ± SD and were analyzed with one-tailed Mann–Whitney test. Per sample, six pictures were randomly taken on two slides at predefined positions (for detailed information see Figure A4, Appendix A) and the number of NET-activated cells were determined. Unstimulated, CD, PMA: *n* = 42 pictures from seven independent experiments; *A.pp*: *n* = 18 pictures from three independent experiments; *S. suis n* = 24 pictures from four independent experiments) (* *p* < 0.05, ** *p* < 0.01, *** *p* < 0.001, **** *p* < 0.0001).

**Figure 2 ijms-23-08953-f002:**
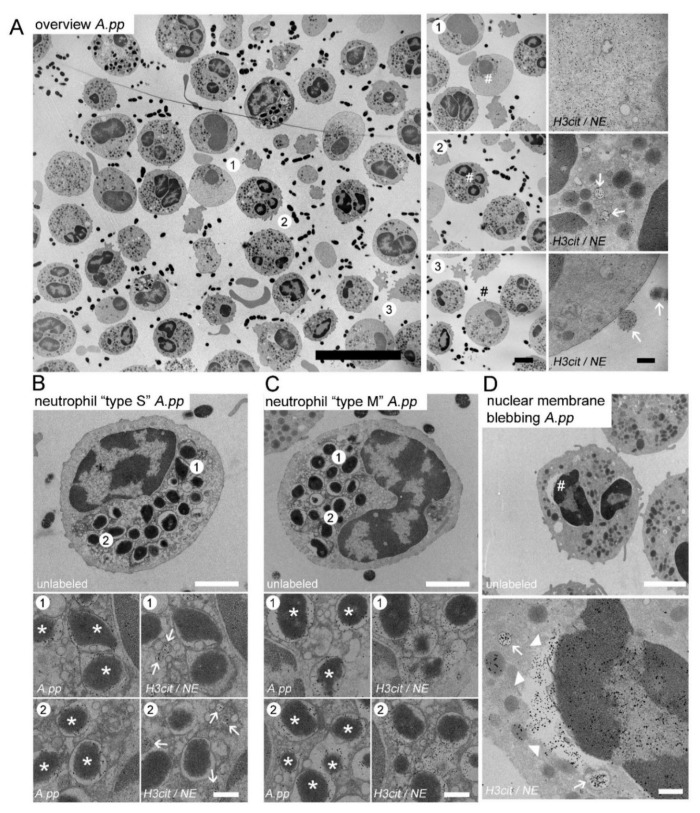
(**A**–**D**) Representative TEM images derived from one experimental run of porcine neutrophils after 30 min of *A.pp* infection; (**A**) Overview picture, (**area 1**) neutrophil containing NE and H3cit in the cytoplasm; (**area 2**) neutrophil containing NE and H3cit in vesicles (white arrows); (**area 3**) neutrophil releasing NE and H3cit positive vesicles in the extracellular space (white arrows). The hash marks the zoom area; (**B**) Neutrophil “type separated = type S”: gold-labeling of two serial sections identified neutrophils with phagocytosed *A.pp* (serial Section 1, left panel, white stars mark *A.pp*) and clearly outside all bacteria-containing phagosomes separated NE/H3cit positive vesicles (right panel, white arrows mark vesicles); (**C**) Neutrophil “type merged = type M”: gold-labeling of two serial sections identified neutrophils with phagocytosed *A.pp* (left panel, white stars mark *A.pp*) and these phagosome are in addition NE/H3cit positive (right panel); (**D**) The hash marks in the upper panel the zoom area. Neutrophils after *A.pp* infection showing nuclear membrane blebbing (NE/H3cit positive gold-labeled, white arrow heads) and vesicles (NE/H3cit, white arrows). Furthermore, NE positive granule and a NE/H3cit positive nucleus were identified; (**A**–**C**) and (serial Section 1, right panels with zoom pictures) and D lower panel: 5 nm gold labeling = H3-cit and 10 nm gold labeling = NE; (**B**,**C**) (serial Section 2, left panels with zoom pictures) 10 nm gold labeling = *A.pp*. Scale bars in **A**: overview = 10 µm; zoom pictures = 2.5 µm; zoom and gold-labeling = 200 nm; (**B–D**) 2 µm (upper panel) and 250 nm (lower panels).

**Figure 3 ijms-23-08953-f003:**
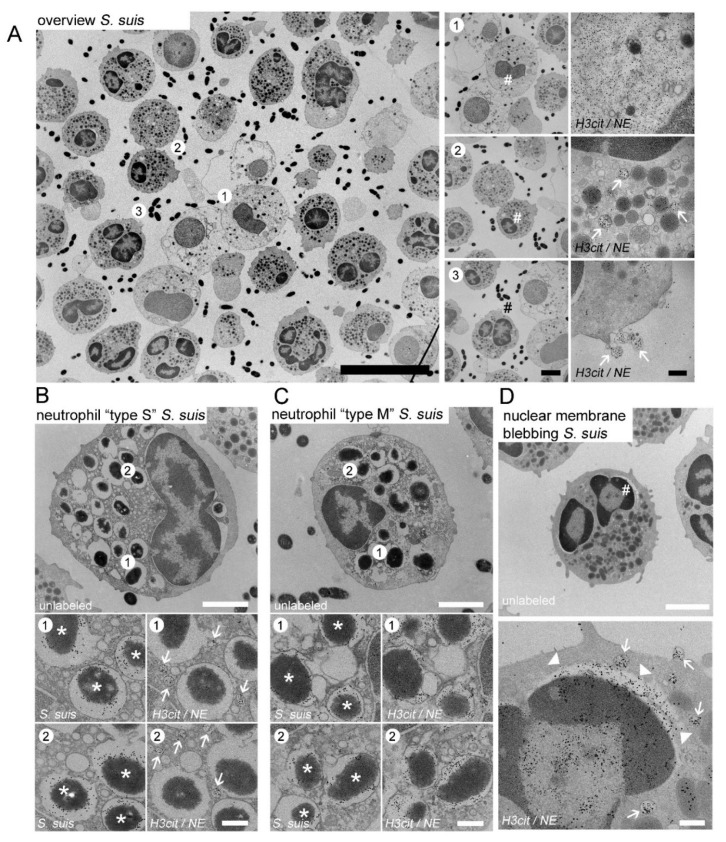
(**A**–**D**) Representative TEM images taken during one experimental run of porcine neutrophils after 30 min of *S. suis* infection; (**A**) Overview picture, (**area 1**) neutrophil containing NE and H3cit in the cytoplasm; (**area 2**) neutrophil containing NE and H3cit in vesicles (white arrows); (**area 3**) neutrophil releasing NE and H3cit positive vesicles in the extracellular space (white arrows); (**B**) Neutrophil “type S”: gold-labeling of two serial sections identified neutrophils with phagocytosed *S. suis* (serial Section 1, left panel, white stars mark *S. suis*) and clearly outside all bacteria-containing phagosomes separated NE/H3cit positive vesicles (serial Section 2, right panel, white arrows mark vesicles); (**C**) Neutrophil “type M”: gold-labeling of two serial sections identified neutrophils with phagocytosed *S. suis* (serial Section 1, left panel, white stars mark *A.pp*) and these phagosome are in addition NE/H3cit positive (serial Section 2, right panel); (**D**) The hash marks in the upper panel the zoom area. Neutrophils after *S. suis* infection showing nuclear membrane blebbing (NE/H3cit positive gold-labeled, white arrow heads) and vesicles (NE/H3cit, white arrows). Furthermore, NE positive granule and a NE/H3cit positive nucleus were identified; (**A**–**C**) (right panels with zoom pictures) and D lower panel: 5 nm gold labeling = H3-cit and 10 nm gold labeling = NE; (**B**,**C**) (left panels with zoom pictures) 10 nm gold labeling = *S. suis*. Scale bars in A: overview = 10 µm; zoom pictures = 2.5 µm; zoom and gold-labeling = 200 nm; (**B**–**D**) 2 µm (upper panel) and 250 nm (lower panels).

**Figure 4 ijms-23-08953-f004:**
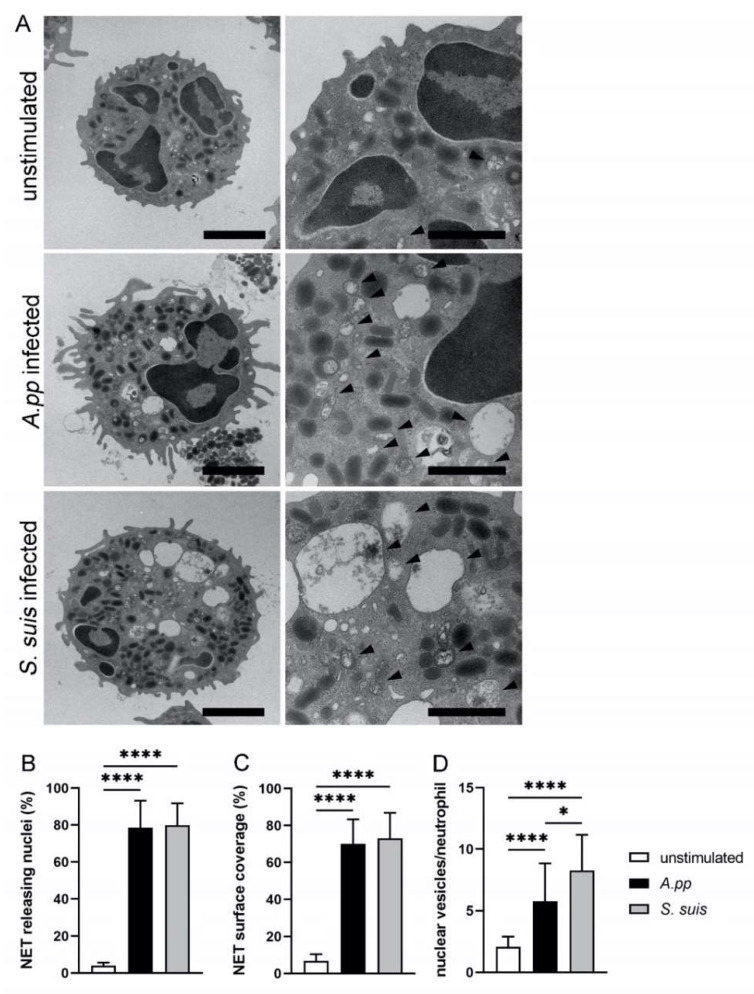
(**A**) Representative TEM images of porcine neutrophils after a 30 min incubation period that were used for quantification (taken from one experimental run). The arrowheads mark vesicles within the neutrophils. The scale bars are: 2 µm (left) and 1 µm (right); (**B**–**D**) Statistical analysis of the TEM images: In total, 30 cellular profiles from randomly selected fields on the thin sections were analyzed per sample. Neutrophils stimulated with *A.pp* and *S. suis* showed a significantly higher NET release from the nuclei (“suicidal NETosis”) (**B**); a significantly higher NET surface coverage (**C**); and a significantly higher number of nuclear vesicles (“vital NETosis”) in neutrophils, with the highest amount in *S. suis* infected neutrophils (**D**). Data were analyzed with the Kruskal–Wallis multiple comparison test (*n* = 30) and presented with mean ± SD (* *p* ≤ 0.05, **** *p* < 0.0001).

## Data Availability

The authors confirm that the data supporting the findings of this study are available within the published article. Raw data were generated at the Department of Biochemistry, University of Veterinary Medicine Hannover, Foundation, Hannover, Germany. Derived data supporting the findings of this study are available from the corresponding authors, N.d.B, on request.

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
