# Peer review of "New Insights into Neutrophil Extracellular Trap (NETs) Formation from Porcine Neutrophils in Response to Bacterial Infections"

_ijms, 2022, doi:10.3390/ijms23168953_

Round 1

Reviewer 1 Report

The manuscript gives evidence of NET formation in porcine neutrophils and provides some insight into possible mechanisms involved. Based on the analysis of fluorescent microscopic images, the authors demonstrate that the treatment of neutrophils with both, a specific gram positive and a specific gram negative bacterium, leads to the formation of NETs. This NET formation can be reduced significantly by DPI, an NADPH oxidase blocker, and in some cases by an anti-CD18a antibody, giving some indication on possible mechanisms involved.

They also provide high quality electron microscopic evidence of suicidal NET formation and of vesicles containing NETs in neutrophils, also supported by convincingly performed post-embedding immunostaining.

However, in my opinion, parts of the experiments and statistical analysis should have been done more thoroughly and some conclusions are not fully supported by the present data (see detailed comments below).

Unfortunately, the quality of writing is poor throughout the manuscript. The context of information given within some paragraphs is not always clear. In parts of the methods-section, processes are described in a confusing manner, relevant information is sometimes missing and less important information is described in great detail. There are numerous spelling errors, many cases of incorrect grammar and inappropriate use of some terms and phrases. Some sentences are hard to understand, mainly due to the appallingly incorrect word order (many examples but not all are pointed out below).

Major concerns:

·         The methods section does not mention that any sera (eg. FBS) have been used in incubation buffers during the experiments. As shown by Kamoshida et al (2017). FEBS Open Bio. 2017;7: 877–886, in vitro neutrophils spontaneously produce NETs in serum-free media. Specifically for the analysis of different mechanisms of NET formation it is important to block spontaneous NETosis in order to exclude various mechanisms being activated.

·         For NETs-quantification, 6 images per specimen were analyzed. In my personal experience, NETs in such samples are often distributed irregularly and clustered, so I am not sure whether 6 images are sufficient for representative photo-sampling. The reference given provides no further information, in fact it is the exact same text as in the present manuscript. It would be helpful if the authors could provide data/low magnification images or similar (or a reference to such data) to corroborate their method of photo-sampling accordingly.

·         As I understand the text, 6 images per samples were quantified and the mean of these 6 images was calculated. These means were then used for further statistical analysis. If this is really the case, I would like to state my concerns about using these calculated means as ’raw-data’ to perform a normality test followed by a t-test, where means of those means are calculated and compared. If I understood this wrong, please rephrase the description in order to avoid such misunderstandings.

·         There is no mention of the assessment of the data presented in Fig 4 anywhere. It is only stated in the results sections that 10 cells per sample have been analyzed without any further information on how these cells were selected. Again, according to my personal experience 10 cells seem an extremely small sample size. Furthermore, there is no description of how the variables quantified were defined, thus, what exactly was counted as `suicidal´ and `vital´ NETs, what is meant by `NETs surface coverage´ and how was it assessed?

·         The authors neglected to state the number of independent repeats of their experiments.

·         In the Methods sections it is stated that due to variations in growth rate the MOI of A. app. was in a range of 0.5 – 2. Has it been tested how this variation influences the quantity of NETs in such assays?

·         Is there any EM evidence of nuclear membrane blebbing, as described by Pilsczek et al. (2010) J.Immunol; 185: 7413–7426.?

·         Fig 2B(1) The vesicle at the top of the right panel indicated as ‘separate vesicle’ is clearly the same compartment containing a bacterium as visible in the adjacent section (left panel). This makes the statement, that in this cell there is no citH3/NE staining in phagolysosomes, invalid.

When interpreting TEM images it must always be kept in mind that it is a 2D section through an irregular 3D structure, taken at a random position. Profiles appearing as separate structures could also be sections of extensions of the same compartment. Accordingly, the structures appearing as separate vesicles could also be parts of the phagosomes. To unambiguously demonstrate that they are separate vesicles, it is necessary to analyze serial sections or at least very large numbers of cells.

·         Has it been considered that the NETs in the phagolysosomes could have been taken up by phagocytosis alongside the bacteria? In Fig 4 it is shown that free NETs are present after 30min incubation with the bacteria, so it seems plausible that these NETs might have adhered to the bacteria and been taken up with them.

Minor points:

·         line 46: loses should be losses

·         Line 50: `Both bacteria induce NETs in pigs´

Reference needed

·         Line 69: `induce thereby´ wrong word order, should be `thereby induce´; `this exotoxins´ should be `these exotoxins´

·         Line 70: please specify `induce a cell signaling´

·         Line 71: incorrect word order

·         Paragraph lines 71 to77: It is not very clear to me how this information fits the context of NETosis

·         Line 87: It’s not the nucleus that decondensates but the chromatin.

·         Lines 91/92: What is meant by `further mechanism´?

·         Lines 92 – 94: There is some mix up with the references, ref 40 refers to the vesicular release of NET.

·         Line 101: Please specify the limitations of NETs quantification by measuring free DNA.

·         Line 122: Please give the full name of `CD ´

·         Line 132: Adverb needed -> remarkably

·         Lines 125, 180, 217: As the cells can be considered to be of the same genotype and were treated similarly, the use of the term `phenotype´ is inappropriate in this context.

·         line 132: `… whereas PMA is not efficiently stimulating porcine neutrophils for NET release´

(1)     Fig 1 shows a significant increase of NETs compared to the unstimulated control. The authors are contradicting their own data.

(2)     Incorrect use of Present Continuous, incorrect word order; it should be 'whereas PMA does not stimulate neutrophils to release NETs efficiently'

·         Fig. 1 B/C: The way the significant differences are indicated is extremely hard to make out. It would be advantageous to find a less confusing way of doing this. Marking ‘almost significant’ differences should be omitted.

·         Lines 160 – 173: I’m not sure why this information is given here, I would rather expect to find it in the introduction.

·         Line 174: `Due to NETs can be induced ROS independent´

This sentence makes no sense

·         Lines 197,198: This information is irrelevant here and can be omitted.

·         Line 202: Why is in vivo - NETosis mentioned here when it has been analyzed in vitro?

·         Line 218: ` and separately vesicles that contain H3cit and NE´ No adverb here, it should be `separate vesicles´.

·         Lines 221, 222: ` Inside “type M” neutrophils partially H3cit and NE can be identified in the cytoplasm.´

This makes no sense to me.

·         Lines 227 ff: Leading finally to the question, at what moment does the neutrophil decide to go on with “vital” NETosis and/or phagocytosis or change the strategy and exclusively perform a “suicidal” NETosis.

(1)     This suggests that there is evidence that such a change in strategy takes place within individual cells. If so, it should be presented in the manuscript. If not, other possibilities should be discussed likewise.

(2)     Incorrect word order and peculiar phasing

·         Lines 234 ff: Nevertheless, our data clearly show that neutrophils after or in parallel undergo phagocytosis of S. suis and A.pp and release vesicular NETs.

(1)     I’m not sure about the `clearly´, see major concerns above.

(2)     Incorrect word order.

·         Line 289: only TEM analysis was used to reliable identify vesicular NETosis

Adverb needed -> reliably

·         Line 292: NET formation is not only existing after bacterial infections

Incorrect use of Present Continuous

·         Line 295: … (eCATH 1) was inducing vesicular NET formation

Incorrect use of Past Continuous

·         Line 458: If the detector is mentioned it would make sense to also mention the microscope used.

·         Line 459: The staining is done before the viewing so the steps should be mentioned in that chronology.

·         Lines 463/464: The secondary antibodies used should be mentioned.

·         Line 465: `70 nm-thick serial sections´

Were the other sections of a different thickness? If not, why mention it here but not above?

·         Line 467/468: This information should be given along with the other antibodies used in lines 461 ff.

·         Lines 492 ff: This sections should be rephrased as, in the present wording, the only conclusion the authors seem to draw from their results is that further studies are needed to investigate vital NETosis after bacterial infection.

·         Line 495: I’m not sure what is meant with `in the absence of antibodies´.

Author Response

Answers to the reviewers IJMS- 1817196 - first report

Dear Editors and Reviewers,

We thank the reviewer’s for the first report and the constructive suggestions.

Please find below our answers to the comments and questions of the reviewers.

We have prepared a revised version of the manuscript, highlighting the changes from first revision using the Track Changes function in Microsoft Word, and we have prepared a point-to-point response in order to address the remarks out by the reviewers.

Based on the comment’s of the two reviewer's we added several results to the Appendix figures (Figures A1, A2, and A4). Furthermore, we rearranged some figures (Figures 1-3, Figure A3). The changed figures are included in new version and uploaded as TIF file.

Based on the comments we rearranged furthermore the complete Introduction, separated Results and Discussion and revised the methods and conclusion part.

We thank again the reviewers for the comments to our study and tried to improve the manuscript based on the constructive comments.

With kind regards

Nicole de Buhr

Reviewer 1

Comments and Suggestions for Authors

The manuscript gives evidence of NET formation in porcine neutrophils and provides some insight into possible mechanisms involved. Based on the analysis of fluorescent microscopic images, the authors demonstrate that the treatment of neutrophils with both, a specific gram positive and a specific gram negative bacterium, leads to the formation of NETs. This NET formation can be reduced significantly by DPI, an NADPH oxidase blocker, and in some cases by an anti-CD18a antibody, giving some indication on possible mechanisms involved.

They also provide high quality electron microscopic evidence of suicidal NET formation and of vesicles containing NETs in neutrophils, also supported by convincingly performed post-embedding immunostaining.

However, in my opinion, parts of the experiments and statistical analysis should have been done more thoroughly and some conclusions are not fully supported by the present data (see detailed comments below).

Unfortunately, the quality of writing is poor throughout the manuscript. The context of information given within some paragraphs is not always clear. In parts of the methods-section, processes are described in a confusing manner, relevant information is sometimes missing and less important information is described in great detail. There are numerous spelling errors, many cases of incorrect grammar and inappropriate use of some terms and phrases. Some sentences are hard to understand, mainly due to the appallingly incorrect word order (many examples but not all are pointed out below).

Answer:

We thank the reviewer for the constructive comments and revised the manuscript accordingly. Please find below a detailed point-to-point response.

Furthermore, we upload a certificate for “Proofreading of attached scientific manuscript” by the English Editorial Office of our university.

Major concerns:

  1. The methods section does not mention that any sera (eg. FBS) have been used in incubation buffers during the experiments. As shown by Kamoshida et al (2017). FEBS Open Bio. 2017;7: 877–886, in vitro neutrophils spontaneously produce NETs in serum-free media. Specifically for the analysis of different mechanisms of NET formation it is important to block spontaneous NETosis in order to exclude various mechanisms being activated.

Answer: We agree with the reviewer that blocking spontaneous NETs would help to better understand the exact mechanism.

However, we have not included FBS into our experiments due to several reasons:

  1. FCS contains nucleases that can influence / degrade NETs as described by one of our co-authors: Fetal calf serum contains heat-stable nucleases that degrade neutrophil extracellular traps. von Köckritz-Blickwede M, Chow OA, Nizet V.Blood. 2009 Dec 10;114(25):5245-6. doi: 10.1182/blood-2009-08-240713.
  2. We are analyzing pathogen-induced NET-formation and cannot exclude an influence of factors from FCS. Therefore, we wanted to exclude ROS-inhibiting effects of FCS, which could then also reduce or increase pathogen-induced NETs. It was already shown by authors for Streptococci that FCS influence virulence factors and growth:

Ref 1: Serotype-specific invasiveness and colonization prevalence in Streptococcus pneumoniae correlate with the lag phase during in vitro growth. Bättig et al. 2006; doi 10.1016/J.MICINF.2006.07.013

Ref 2: Human Serum Induces Streptococcal C5a Peptidase Expression. Gleich-Theurer et al. 2009; Doi: 10.1128/IAI.00826-08

  1. We conducted a test experiment based on the publication Kamoshida et al (2017) and compared NET induction (3 h) in the presence and absence of 2% FBS (heat inactivated). We could not observe a decreased amount of NETs in an untreated (negative) control compared to a sample without FCS. Furthermore, we observed with the porcine neutrophils another phenotype, as the FBS treated neutrophils showed a tendency to clump. We included a new Figure in the Appendix Figure A2 with representative images.

Furthermore, we include in the results part the following paragraph:

Lines 128 ff:

“Secondly, we investigated whether the addition of 2 % fetal bovine serum (FBS) reduces spontaneous NET release in porcine neutrophils as described by Kamoshida et al.[56]. A reduction in spontaneous NETs could help to identify small differences after treatment. Compared to a sample without FBS, we did not identify by qualitative analysis fewer NETs in untreated neutrophils in the presence of FBS, but instead observed another critical phenomenon. Porcine neutrophils that were incubated with FBS showed a tendency to clump together (Figure A2). Therefore, all following assays were conducted in the absence of FBS.”

  1. For NETs-quantification, 6 images per specimen were analyzed. In my personal experience, NETs in such samples are often distributed irregularly and clustered, so I am not sure whether 6 images are sufficient for representative photo-sampling. The reference given provides no further information, in fact it is the exact same text as in the present manuscript. It would be helpful if the authors could provide data/low magnification images or similar (or a reference to such data) to corroborate their method of photo-sampling accordingly.

 Answer: We agree that the counting method for NETs of confocal microscopy pictures needs a well-established protocol to exclude effects on the phenotypes by an irregularly distribution of cells or a clustering of neutrophils. Based on our experience of NET analysis since more than 10 years, we established in our lab this quantification method following a SOP for the counting. The whole analysis starts with well-established SOPs for the Poly-L-Lysine counting, species specific neutrophil isolation, seeding procedure, NET induction and staining followed by a confocal microscopy analysis by intensively trained researchers. This training includes a training for counting to ensure a high quality of counting results and the possibility to compare results that are analyzed by different researchers in our group over years. To clarify how the counting was conducted, we included a new figure in the Appendix (Figure A4). Here we compared two counting methods (counting the total number of cells on six images that were taken on different positions of the same sample).

We identified that the number of cells per six images does not differ in the mean between imaging method number 1 and 2. However, we prefer to use two slides as this includes a technical replicate in the analysis.

To clarify the counting method, we extended the methods section as follows (lines 518 ff):

“The pictures were taken at predefined positions on two slides (see Figure A4). Only if an artefact (e.g. air bubble) was present was another adjacent area selected. The focus was set on the nuclei (blue channel). The cells were counted manually using ImageJ software (version 1.52q, National Institute of Health, USA) with the Cell counter plugin. All NET negative and positive neutrophils were counted. Neutrophils on the border of an image (not fully visible) were not counted at all. In each sample, a minimum of 300 cells were counted. A neutrophil was counted as positive if an evident off-shoot of DNA was visible or if at least two of the following criteria were found: enlarged nucleus, decondensed nucleus or blurry rim. The percentage of NET-positive neutrophils was calculated (NET-activated cells). An average from the counted NET-activated cells from the six pictures from each sample was calculated.”

With a lower magnification (20x instead of 40x objective with oil) it would be not possible to our experience to characterize the neutrophils for NET activation / release.

  1. As I understand the text, 6 images per samples were quantified and the mean of these 6 images was calculated. These means were then used for further statistical analysis. If this is really the case, I would like to state my concerns about using these calculated means as ’raw-data’ to perform a normality test followed by a t-test, where means of those means are calculated and compared. If I understood this wrong, please rephrase the description in order to avoid such misunderstandings.

Answer: Based on this question and stated concerns, we have recalculated the statistics as following: In each experimental run we have randomly taken six pictures and counted the NET positive cells as described more in detail in point 2 (see above). Therefore, from each run we included all results into one statistical analysis and calculated the normality test (Shapiro-Wilk test). As not all samples passed the normality test (alpha=0.05), we analyzed in the next step in each stimulus group the statistical differences with Mann-Whitney test and accordingly updated the Figure 1.

Therefore, the figure legend was changed as follows (lines 179 ff):

(B) Statistical analysis of NET inhibition assay (3h stimulation) sorted by stimulus. NET release by A.pp is significantly blocked by antiCD18a and DPI. NET release by S. suis is significantly blocked by DPI and significantly increased by antiCD18a co-incubated with S. suis. Data are presented with mean ± SD and were analyzed with one-tailed Mann Whitney test. Per sample, six pictures were randomly taken on two slides at predefined positions (for detailed information see Figure A4) and the number of NET-activated cells were determined. Unstimulated, CD, PMA: n = 42 pictures from seven independent experiments; A.pp: n = 18 pictures from three independent experiments; S. suis n = 24 pictures from four independent experiments) (* p < 0.05, ** p < 0.01, *** p < 0.001, p **** <0.0001).”

  1. There is no mention of the assessment of the data presented in Fig 4 anywhere. It is only stated in the results sections that 10 cells per sample have been analyzed without any further information on how these cells were selected. Again, according to my personal experience 10 cells seem an extremely small sample size. Furthermore, there is no description of how the variables quantified were defined, thus, what exactly was counted as `suicidal´ and `vital´ NETs, what is meant by `NETs surface coverage´ and how was it assessed?

 Answer: We apologize this confusing information in the figure legend and corrected the information as follows (lines 276):

“In total, 30 cellular profiles from randomly selected fields on the thin sections were analyzed per sample.”

A detailed information is now presented in the material and methods section and hopefully clarify that much more than 10 cells were counted:

Regarding the second part of this comment, we included more information into the results and discussion:

Lines 261 ff

“Neutrophils that showed a loss of intact cytoplasmatic structure with release of nuclear DNA were counted as neutrophils undergoing “suicidal” NETosis.”

Lines 265 ff

“Furthermore, A.pp and S. suis significantly induced higher numbers of vesicles (characteristic of viable / vesicular NETosis) per neutrophil compared to an unstimulated control.”

Lines 324 ff

“When comparing “suicidal” and “vital” NETosis in our investigated samples, we identified that both pathogens induced “suicidal” NETosis (NET release from the nuclei, Figure 4B) to comparable amounts after 30 minutes infection with an MOI of 10.”

Furthermore, we included into the Material and methods section the following information about the quantification method (lines 559 ff):

4.9 NET quantification in EM images

            NETs were quantified as follows: in total, cellular profiles from 30 randomly selected fields on the thin sections were analyzed per sample for three different analysis points. For each analysis point, 30 different fields were assessed as follows:

  1. NET release from the nuclei: loss of intact cytoplasmatic structure with release of nuclear DNA was counted as positive. The percentage of positive cells was calculated in relation to all observed cells investigated.
  2. NET surface coverage: area covered with NET structures that were found in the extracellular environment of neutrophils was measured. The percentage indicated the fraction of the area covered by NETs in relation to the total area of the respective field.
  3. Nuclear vesicles per neutrophil: the number of intracellular nuclear vesicles (H3cit and NE positive) were counted per neutrophil in randomly selected fields. The analyzed neutrophils show an intact cytoplasmic structure with nucleus, double nuclear membrane, and intact granules.”
  4. The authors neglected to state the number of independent repeats of their experiments.

 Answer: In Figure 1 this information was already included and is now updated based on the new Fig. 1:

“Unstimulated, CD, PMA: n = 42 pictures from seven independent experiments; A.pp: n = 18 pictures from three independent experiments; S. suis n = 24 pictures from four independent experiments) (* p < 0.05, ** p < 0.01, *** p < 0.001, p **** <0.0001).”

In figure 2-4 we have only conducted one independent experimental run. We included therefore in all figure legends the missing information:

“derived from one experimental run”

As the analysis by TEM is expensive and time-consuming (compared to other visualization methods) we have conducted only the analysis of one experimental run. However, we discuss this issue now in the discussion as follows (lines 358 ff):

“Therefore, an observation of individual neutrophils with live cell imaging techniques would be needed that can visualize vesicles and bacteria inside a neutrophil. As live cell imaging with TEM is not yet possible [62], the only option would be high resolution immunofluorescence microscopy. Nonetheless, there are several technical challenges that need to be solved: 1. The assays needs to be performed with fluorescence labeled bacteria and the influence of this labeling on phagocytosis and NETosis needs to be investigated first. 2. The vesicles we observed in this study have a mean size of only 200-300 nm. 3. A specific marker for vesicular NETs (H3cit and NE positive vesicles) working in live-cell imaging needs to be found and established. The study by Thiam et al. characterized NETosis in human and murine neutrophils and differentiated HL-60 neutrophil-like cells (dHL-60) by analyzing plasma membrane microvesicles with high-resolution time-lapse microscopy [63]. The authors observed that NETosis took place in a stepwise manner, and they showed that the yeast Candida albicans was actively phagocytosed by neutrophils and these neutrophils additionally completed NETosis quickly. It was described that neutrophils can sense the size of a microbe and for this reason can undergo fast phagocytosis to encounter small pathogens like bacteria or slow “suicidal” NETosis to fight against large pathogens like Candida albicans hyphae [64].”

  1. In the Methods sections it is stated that due to variations in growth rate the MOI of  app.was in a range of 0.5 – 2. Has it been tested how this variation influences the quantity of NETs in such assays?

 Answer: We have performed all assays with freshly grown A.pp as it was already published by us (Degraded neutrophil extracellular traps promote the growth of Actinobacillus pleuropneumoniae, de Buhr et al. 2019). In this previously published work, we used the same method and did not observe a high standard deviation between different experiments, indicating less influence by the different MOIs. In this recent manuscript we conducted the NET induction assays presented in Figure 1 with the same method. Nevertheless, we agree that one fixed MOI could reduce the risk for higher standard deviations. Therefore, we worked during the process of this manuscript (started in 2019) to establish a protocol for working cryostocks that can be used from specific growth phases of A.pp (as we have it since long time established for S. suis). However, A.pp is fragile in this cryo conservation process and we had to run several trials. Now, we have a protocol established that allows us to use working cryostocks of A.pp (as used for the EM analysis) and therefore we can infect with a specific MOI.

Therefore, we included now a new Figure A1 into the appendix that shows a quantitative overview of neutrophils infected for 3 hours with A.pp MOI 0.5, 1, and 2.

We included into the results part the following paragraph (lines 124 ff):

“Firstly, we conducted screening assays to adjust the experimental settings. Different infection doses of A.pp (multiplicity of infection (MOI) of 0.5, 1 and 2) induced NET formation after 3 hours and no difference was observed (Figure A1). Therefore, a range of MOI = 0.5-2 for A.pp (freshly grown) was used in the following assays.”

  1. Is there any EM evidence of nuclear membrane blebbing, as described by Pilsczek et al. (2010) J.Immunol; 185: 7413–7426.?

 Answer: Yes, this can occasionally be seen, too, but nuclear vesicles present in the cytoplasm, apparently already detached from the nucleus, are more frequent in these samples. Pilsczek et al. stated in their article that NETosis is dependent on the bacterial species and they had made their observations with Staph. aureus infection so there could actually be slightly different NETosis mechanisms here. However, future studies would be needed to investigate this for S. suis and A.pp more in detail. We included example images in Figs. 2D and 3D and paragraphs into the results and discussion:

Lines 213 ff

“Furthermore, we identified occasionally nuclear membrane blebbing positive for H3cit and NE (Figure 2D and 3D), however nuclear vesicles present in the cytoplasm, apparently already detached from the nucleus or released from the neutrophil, were more frequently identified in these samples.”

Lines 316 ff

“In the study by Pilsczek et al. a nuclear membrane blebbing was observed with TEM images. This describes a separation of the inner nuclear membrane from the outer nuclear membrane. Inside the space between these two membranes, DNA with nucleosomes were observed, referred to as “beads on a string”. Indeed, we also occasionally observed in our samples after being infected with S. suis and A.pp for 30 minutes, a separation of the two nuclear membranes. However, nuclear vesicles present in the cytoplasm, apparently already detached from the nucleus, occurred more frequently in our samples (Figs. 2D and 3D).

  1. Fig 2B(1) The vesicle at the top of the right panel indicated as ‘separate vesicle’ is clearly the same compartment containing a bacterium as visible in the adjacent section (left panel). This makes the statement, that in this cell there is no citH3/NE staining in phagolysosomes, invalid.

Answer: We did not state that a neutrophil cannot contain separate vesicles and phagolysomes. However, we describe that in neutrophils of the separate type we identify with the serial sections (section 1 always left side and stained for H3cit /NE and section 2 always right side stained for A.pp or S. suis) a separate vesicles inside the neutrophil (H3cit /NE) and phagolysosomes positive only for the bacteria. The described and questioned figure 2B (right panel) are clearly outside all bacteria-containing phagosomes, and this is also true for the left panel in the adjacent serial section. Therefore, we show the other “type M”, where we identified with the same technique of serial sections phagosomes that are positive for bacteria (left panel, white star marks them) and in section 2 the bacteria-containing phagosomes are stained for H3cit /NE (right panel).

To clarify this for the reader, we have rewritten the legends of Figure 2 and 3 (lines 219 ff and lines 240 ff):

“Neutrophil “type separated = type S”: gold-labeling of two serial sections identified neutrophils with phagocytosed A.pp (serial section 1, left panel, white stars mark A.pp) and clearly outside all bacteria-containing phagosomes separated NE / H3cit positive vesicles (right panel, white arrows mark vesicles). (C) Neutrophil “type merged = type M”: gold-labeling of two serial sections identified neutrophils with phagocytosed A.pp (left panel, white stars mark A.pp) and these phagosome are in addition NE / H3cit positive (right panel). (D) The hash marks in the upper panel the zoom area. Neutrophil after A.pp infection showing nuclear membrane blebbing (NE / H3cit positive gold-labeled, white arrow heads) and vesicles (NE / H3cit, white arrows). Furthermore, NE positive granule and a NE / H3cit positive nucleus were identified. A and B-C (serial section 1, right panels with zoom pictures) and D lower panel: 5 nm gold labeling= H3-cit and 10 nm gold labeling = NE. B-C (serial section 2, left panels with zoom pictures) 10 nm gold labeling = A.pp. Scale bars in A: overview= 10 µm, zoom pictures = 2.5 µm, zoom and gold-labeling = 200 nm; B-D 2 µm (upper panel) and 250 nm (lower panels).”

Equal adaption in legend of Figure 3.

  1. When interpreting TEM images it must always be kept in mind that it is a 2D section through an irregular 3D structure, taken at a random position. Profiles appearing as separate structures could also be sections of extensions of the same compartment. Accordingly, the structures appearing as separate vesicles could also be parts of the phagosomes. To unambiguously demonstrate that they are separate vesicles, it is necessary to analyze serial sections or at least very large numbers of cells.

 Answer: This is a very good point from the reviewer! To tackle this common problem in thin sectioning and TEM, 30 cellular profiles from random locations on the thin sections are assessed, ensuring that as much 3-D information as possible is accounted for in the evaluations. This is a common procedure in electron microscopy in order to avoid extensive time- and money-consuming 3D reconstructions of serial sections. We included the informations about the analysis method in the material and methods part as described above.

  1. Has it been considered that the NETs in the phagolysosomes could have been taken up by phagocytosis alongside the bacteria? In Fig 4 it is shown that free NETs are present after 30min incubation with the bacteria, so it seems plausible that these NETs might have adhered to the bacteria and been taken up with them.

 Answer: We agree with the reviewer that this process would be possible. Therefore, we added the following sentence to the discussion (lines 381 ff):

“Our data do not reveal whether these neutrophils phagocytosed already before bacteria or whether bacteria that where extracellular entrapped in NETs were phagocytosed together with the NETs structures. This could also explain the “type M” neutrophils.”

Minor points:

  • line 46: loses should be losses (Changed)
  • Line 50: `Both bacteria induce NETs in pigs´
    • Reference needed

We added:

  • de Buhr, N.; Neumann, A.; Jerjomiceva, N.; von Köckritz-Blickwede, M.; Baums, C.G. Streptococcus suis DNase SsnA contributes to degradation of neutrophil extracellular traps (NETs) and evasion of NET-mediated antimicrobial activity. Microbiology 2014, 160, 385–395.
  • de Buhr, N.; Bonilla, M.C.; Pfeiffer, J.; Akhdar, S.; Schwennen, C.; Kahl, B.C.; Waldmann, K.; Valentin-Weigand, P.; Hennig-Pauka, I.; von Köckritz-Blickwede, M. Degraded neutrophil extracellular traps promote the growth of Actinobacillus pleuropneumoniae. Cell Death Dis. 2019, 10, 657.
  • Ma, F.; Chang, X.; Wang, G.; Zhou, H.; Ma, Z.; Lin, H.; Fan, H. Streptococcus Suis Serotype 2 Stimulates Neutrophil Extracellular Traps Formation via Activation of p38 MAPK and ERK1/2. Front. Immunol. 2018, 9, 2854.

  • Line 69: `induce thereby´ wrong word order, should be `thereby induce´; `this exotoxins´ should be `these exotoxins´ (changed)
  • Line 70: please specify `induce a cell signaling´ è due to rewriting of the introduction, this part was deleted.

  • Line 71: incorrect word order:

Answer: the sentence was rewritten as follows “A.pp produces pore-forming exotoxins (Apx toxins) as one of the most important virulence factors, which, like LKT, belong to the family Repeats in ToXin (RTX) [38].

  • Paragraph lines 71 to77: It is not very clear to me how this information fits the context of NETosis

Answer: We have rewritten this paragraph

  • Line 87: It’s not the nucleus that decondensates but the chromatin. (Changed)
  • Lines 91/92: What is meant by `further mechanism´?

Answer: Changed to: “This mechanism is described as an antimicrobial cell death, as the neutrophil dies, releases antimicrobial substances but cannot perform other defense mechanisms like phagocytosis or transmigration [12,13].”

  • Lines 92 – 94: There is some mix up with the references, ref 40 refers to the vesicular release of NET.

Answer: we have corrected the mix up of references.

  • Line 101: Please specify the limitations of NETs quantification by measuring free DNA.

Answer: we added the following informations “However, the quantification of free DNA as NET marker has some limitations; as for example, it is not possible to differentiate between necrosis and NETosis. Furthermore, the frequently used PicoGreen assay to detect free DNA lacks sensitivity to measure a relatively small amount of NETs [52].”

  • Line 122: Please give the full name of `CD ´.

Answer we added: “methyl-β-cyclodextrin”

  • Line 132: Adverb needed -> remarkably

Answer: we have rewritten this sentence and shifted to the discussion.

  • Lines 125, 180, 217: As the cells can be considered to be of the same genotype and were treated similarly, the use of the term `phenotype´ is inappropriate in this context.

 Answer: These lines were changed accordingly, and the term “phenotype” was changed in the entire manuscript.

  • line 132: `… whereas PMA is not efficiently stimulating porcine neutrophils for NET release´

(1)     Fig 1 shows a significant increase of NETs compared to the unstimulated control. The authors are contradicting their own data.

(2)     Incorrect use of Present Continuous, incorrect word order; it should be 'whereas PMA does not stimulate neutrophils to release NETs efficiently'

 Answer: We agree with the reviewer and have deleted this part from the manuscript.

Fig. 1 B/C: The way the significant differences are indicated is extremely hard to make out. It would be advantageous to find a less confusing way of doing this. Marking ‘almost significant’ differences should be omitted.

Answer:

We have changed Fig.1 (NET induction assay) and Fig A3 (Pico Green assay) and present only significant differences <0.05.

  • Lines 160 – 173: I’m not sure why this information is given here, I would rather expect to find it in the introduction.

 Answer: We agree with the reviewer and integrated these parts of this paragraph into the introduction.

Line 174: `Due to NETs can be induced ROS independent´

This sentence makes no sense

Answer: We have deleted this part.

  • Lines 197,198: This information is irrelevant here and can be omitted.

 Answer: We have deleted this part from the results part.

Line 202: Why is in vivo - NETosis mentioned here when it has been analyzed in vitro?

 Answer: we have deleted this part from the results part and included parts in the discussion.

  • Line 218: ` and separately vesicles that contain H3cit and NE´ No adverb here, it should be `separate vesicles´.

 Answer: we changed it accordingly.

  • Lines 221, 222: ` Inside “type M” neutrophils partially H3cit and NE can be identified in the cytoplasm.´

This makes no sense to me.

 Answer: We have deleted this part from the manuscript.

- Lines 227 ff: Leading finally to the question, at what moment does the neutrophil decide to go on with “vital” NETosis and/or phagocytosis or change the strategy and exclusively perform a “suicidal” NETosis.

(1)     This suggests that there is evidence that such a change in strategy takes place within individual cells. If so, it should be presented in the manuscript. If not, other possibilities should be discussed likewise.

(2)     Incorrect word order and peculiar phasing

 Answer: We have rewritten this part and included into the discussion as following:

Lines 355 ff

“Future studies should focus on answering the question at what moment a neutrophil or a specific neutrophil population can undergo “vital” NETosis and/or phagocytosis or change the strategy and exclusively perform “suicidal” NETosis. One other possibility is that every neutrophil can undergo all mechanisms and it is only a question at which time-point the neutrophils was observed.”

  • Lines 234 ff: Nevertheless, our data clearly show that neutrophils after or in parallel undergo phagocytosis of S. suis and A.pp and release vesicular NETs.

(1)     I’m not sure about the `clearly´, see major concerns above.

(2)     Incorrect word order.

Answer: we have explained above the major concern and rewritten this part in the discussion.

  • Line 289: only TEM analysis was used to reliable identify vesicular NETosis

Adverb needed -> reliably (Changed)

  • - Line 292: NET formation is not only existing after bacterial infections

Incorrect use of Present Continuous

Answer: We have rewritten the sentence as follows:

“Interestingly, vesicular NET formation could not only be detected after bacterial infections, but it was also detected ex vivo by our research group in neutrophils from vitreous body fluid of a horse with an inflammatory eye disease (equine recurrent uveitis) [58].”

Line 295: … (eCATH 1) was inducing vesicular NET formation

Incorrect use of Past Continuous

Answer: this part was corrected “(eCATH 1) induced vesicular NET formation.”

Line 458: If the detector is mentioned it would make sense to also mention the microscope used.

 Answer: we agree and included the following information (lines 559 ff) “Images were recorded using a Philips/FEI CM100 BioTwin transmission electron microscope operated at a 60-kV accelerating voltage with a Gatan Multiscan 791 charge-coupled device camera.

Line 459: The staining is done before the viewing so the steps should be mentioned in that chronology.

Answer: We agree and have changed this part accordingly.

  • Lines 463/464: The secondary antibodies used should be mentioned.

 Answer: We included the missing informations.

  • Line 465: `70 nm-thick serial sections´

Were the other sections of a different thickness? If not, why mention it here but not above?

 Answer: All section, including non serial sections, are 70 nm-thick. Therefore, we included the following information in this paragraph “Sections were cut in 70 nm slices with an LKB ultratome and mounted on Formvar-coated copper grids.”

  • Line 467/468: This information should be given along with the other antibodies used in lines 461 ff.

 Answer: This paragraph was changed accordingly.

Lines 492 ff: This sections should be rephrased as, in the present wording, the only conclusion the authors seem to draw from their results is that further studies are needed to investigate vital NETosis after bacterial infection.

 Answer: We have rewritten the conclusion (see lines 600 ff):

“The interaction of porcine neutrophils and Gram positive (S. suis) and Gram negative (A.pp) bacteria is an important host-pathogen interaction in the early phase of an infection.  We observed that porcine neutrophils already after 30 minutes of infection with S. suis serotype 2 strain 10 and A.pp serotype 2 counteract the infection by phagocytosis and in parallel vesicular NETosis (“vital” NETs). Furthermore, other neutrophils already performed at the same time “suicidal” NETosis, which is characterized by NET release from the nuclei and NET structures that cover the surrounding surface of neutrophils. In general, our data show that porcine neutrophils from one donor react to porcine pathogens in a short time with more than one defense mechanism. Interestingly, some differences in the number of vesicles per neutrophil were identified depending on the infecting bacteria. This indicates that neutrophils do not react entirely in the same way with an “all in” reaction after bacterial infection and that a bacteria-specific reaction of the neutrophils is triggered.”

Line 495: I’m not sure what is meant with `in the absence of antibodies´.

Answer: We have deleted this part.

Reviewer 2 Report

This is a carefully documented investigation of an important question.

Introduction

Please restructure your introduction into the key points you wish to make then within each section structure the paragraphs so that one paragraph leads logically to the next.

Results.

Please place your interpretations and queries in the discussion section where you can assemble the evidence pro and contra for each point you make.

These are small queries:

125 this phenotype was clearer after 3h.[please clarify]

127 DPI was leading to the strongest inhibition. [please provided quantitative documentation]

131 highly induces NETs [please quantify]

132 not efficiently stimulating porcine neutrophils [please quantify]

174 Due to NETs can be induced ROS independent, [please clarify]

180 opposite phenotype [please describe and clarify]

Methods

345 (the strain was identified ) “ …from the lung tissue of a diseased fattening pig during an A.pp 345 outbreak”  [This implies you had only one sample. This is key and should be clearly stated in the Abstract and Conclusions sections].

354 “For preparing cryostocks from Streptococcus (S.) suis cps type 2 strain 10, kindly provided by Hilde Smith (Wageningen, GE, The Netherlands)” [again, this implies you had only one sample. This is key and should be clearly stated in the Abstract and Conclusions sections].

Conclusion

The conclusion is very short. Please clarify the fact that you had only on sample for each bacterium although you tested it in multiple well plates. Please present the types of neutrophil responses and clarify quantitatively the responses you found, and the generalisability of your findings and the interpretations you made to guide future researchers  

The English text is exemplary. There are three small typos.

85 oxid to s [change to oxide]

123 remarkable less [change to remarkably]

289 to reliable identify [change to reliably

Author Response

Answers to the reviewers IJMS- 1817196 - first report

Dear Editors and Reviewers,

We thank the reviewer’s for the first report and the constructive suggestions.

Please find below our answers to the comments and questions of the reviewers.

We have prepared a revised version of the manuscript, highlighting the changes from first revision using the Track Changes function in Microsoft Word, and we have prepared a point-to-point response in order to address the remarks out by the reviewers.

Based on the comment’s of the two reviewer's we added several results to the Appendix figures (Figures A1, A2, and A4). Furthermore, we rearranged some figures (Figures 1-3, Figure A3). The changed figures are included in new version and uploaded as TIF file.

Based on the comments we rearranged furthermore the complete Introduction, separated Results and Discussion and revised the methods and conclusion part.

We thank again the reviewers for the comments to our study and tried to improve the manuscript based on the constructive comments.

With kind regards

Nicole de Buhr

Reviewer 2:

Comments and Suggestions for Authors

This is a carefully documented investigation of an important question.

  1. Introduction: Please restructure your introduction into the key points you wish to make then within each section structure the paragraphs so that one paragraph leads logically to the next.

Answer: We thank the reviewer for this comment and have rewritten the introduction into the key points: NETs in general including vital and suicidal NETosis, description of pathogens, NET induction by toxins and toxins in the investigated pathogens and finally techniques to investigate NETs as well as the aim of the study.

We hope that this new structure helps to better understand the content.

  1. Results: Please place your interpretations and queries in the discussion section where you can assemble the evidence pro and contra for each point you make.

Answer: Based on this comment, we have restructured the manuscript into one Results part (2. Results) and a Discussion (3. Discussion). The discussion was extended including some answers to questions from reviewer 1.

These are small queries:

  1. 125 this phenotype was clearer after 3h.[please clarify]

127 DPI was leading to the strongest inhibition. [please provided quantitative documentation]

131 highly induces NETs [please quantify]

132 not efficiently stimulating porcine neutrophils [please quantify]

Answer: As written above, we have restructured the combined results and discussion part into a separated results and discussion section. In addition, we have rewritten the part lines 115 ff. and clarified the four questions/points above.

New sentence for old line 125: “This difference was significant after 3 h DPI treatment and A.pp infection (3 h DPI = 0.74 µg/mL, p= 0.002).”

New sentence for old line 127: “Neutrophils significantly released fewer NETs after A.pp and S. suis infection in the presence of DPI compared to samples without an inhibitor (DPI: A.pp: p<0.0001 and S. suis: p<0.0001) (Figure 1B).”

  1. 174 Due to NETs can be induced ROS independent, [please clarify]

Answer: We have rewritten this part as follows “Furthermore, we investigated the role of CD18 activation in NET formation. A.pp induced CD18-dependent NETs (Figure 1B, p=0.0085).Additionally, a significant difference was observed between anti CD18a and DPI pretreated and A.pp infected neutrophils (p=0.0218) (Figure 1B).”

  1. 180 opposite phenotype [please describe and clarify]

Answer: This part was rewritten as follows:

“Whereas antiCD18a partially blocked NET release by A.pp, another result was observed after S. suis infection of antiCD18a incubated neutrophils. Surprisingly, the pre incubation with antiCD18a significantly increased the NET release by S. suis-infected neutrophils (p=0.0283), indicating that S. suis did not induce NETs via CD18. A significant difference was observed between antiCD18a and DPI pretreated and S. suis infected neutrophils (p<0.0001) (Figure 1B).”

  1. Methods 345 (the strain was identified ) “ …from the lung tissue of a diseased fattening pig during an A.pp 345 outbreak”  [This implies you had only one sample. This is key and should be clearly stated in the Abstract and Conclusions sections].

Answer: Thank you for this comment. We included the information in the abstract and have rewritten the methods section to clarify this question as following:

Line 23 “The aim of this study was to investigate whether A.pp (serotype 2, C3656/0271/11) and S. suis (serotype 2, strain 10) induce NETs by NADPH oxidase or CD18-dependent mechanisms and to characterize phenotypes of NETs in porcine neutrophils.”

Lines 34 ff “In conclusion, both pathogens induce ROS-dependent NETs. Vesicular NETosis and phagocytosis occur in parallel in porcine neutrophils in response to S. suis serotype 2 and A.pp serotype 2.

Indeed, we used only one serotype of A.pp. However, the bacteria were grown fresh for all assays. The streak outs were conducted from one original cyrostock from the strain collection. Detailed informations are now included in the material section:

Lines 410 ff

“ 4.2 Growth conditions of A.pp and S. suis

In this study A.pp serotype (ST) 2 C3656/0271/11 was used. This strain was isolated during routine diagnostics at the Institute of Microbiology, University of Veterinary Medicine Hannover, Foundation, from the lung tissue of a diseased fattening pig during an A.pp outbreak [67]. The isolated strain was stored in the strain collection at -80 °C. A streak out was made on boiled blood agar plates with nicotinamidadenindinucleotide (NAD) and incubated for 24 h at 37 °C with 5% CO2. The streak out was stored at 4 °C and used for a maximum period of 7 days. A.pp ST 2 was grown in medium as described previously [25]. Briefly, bacteria were grown in pleuropneumonia like organism medium (PPLO) supplemented with 0.1% Tween 80 and Isovitale X to the late exponential growth phase. This growth phase is characterized by an optical density (OD600nm) of 0.60±0.01. Bacteria were washed twice in 1× PBS (Lipopolysaccharides (LPS) free). Fresh grown A.pp was used in the NET inhibition assay (Figure 1 and Figure A3). The pellet was finally adjusted to an OD600nm = 0.6 with 1× PBS LPS free and used fresh. In all other presented data, bacteria were grown to late exponential growth phase OD600nm = 0.60±0.01, washed with 1× PBS and frozen in liquid nitrogen and stored at -80 °C until usage (production of working cryo stocks). A thawed working cryostock was used only once.

Streptococcus (S.) suis cps type 2 strain 10, kindly provided by Hilde Smith (Wageningen, GE, the Netherlands) [68], was stored in the strain collection at -80 °C. A streak out was made on Columbia agar with 5 % sheep blood (Merck GmbH, Darmstadt, Germany) and incubated for 24 h at 37 °C. The streak out was stored at 4 °C and used for a maximum period of 7 days.

For preparing working cryostocks, bacteria were freshly grown in tryptic soy broth (TSB) without dextrose (Becton Dickinson, 286220, Franklin Lakes, NJ, USA) until the early stationary growth phase as described before [69]. The culture was mixed with glycerol (final concentration of 15%) and aliquots were immediately frozen in liquid nitrogen and stored at -80 °C until usage. Each working cryostock was used only once.”

  1. 354 “For preparing cryostocks from Streptococcus (S.) suis cps type 2 strain 10, kindly provided by Hilde Smith (Wageningen, GE, The Netherlands)” [again, this implies you had only one sample. This is key and should be clearly stated in the Abstract and Conclusions sections].

Answer: Thank you for this comment. We included the information in the abstract and have rewritten the methods section to clarify this (see above). We used one isolate that was multiplied and working cryo stocks were produced. Indeed, we used only one serotype of S. suis. The streak outs were conducted from one original cyrostock from the strain collection. For each assay prepared working cryostocks were thawed and used only once.

  1. Conclusion

The conclusion is very short. Please clarify the fact that you had only on sample for each bacterium although you tested it in multiple well plates. Please present the types of neutrophil responses and clarify quantitatively the responses you found, and the generalisability of your findings and the interpretations you made to guide future researchers  

Answer: We thank the reviewer for this comment. We have rewritten the conclusion based on this question and the question by reviewer 1 as follows:

The interaction of porcine neutrophils and Gram positive (S. suis) and Gram negative (A.pp) bacteria is an important host-pathogen interaction in the early phase of an infection.  We observed that porcine neutrophils already after 30 minutes of infection with S. suis serotype 2 strain 10 and A.pp serotype 2 counteract the infection by phagocytosis and in parallel vesicular NETosis (“vital” NETs). Furthermore, other neutrophils already performed at the same time “suicidal” NETosis, which is characterized by NET release from the nuclei and NET structures that cover the surrounding surface of neutrophils. In general, our data show that porcine neutrophils from one donor react to porcine pathogens in a short time with more than one defense mechanism. Interestingly, some differences in the number of vesicles per neutrophil were identified depending on the infecting bacteria. This indicates that neutrophils do not react entirely in the same way with an “all in” reaction after bacterial infection and that a bacteria-specific reaction of the neutrophils is triggered.

  1. The English text is exemplary. There are three small typos.

85 oxid to s [change to oxide]

123 remarkable less [change to remarkably]

289 to reliable identify [change to reliably

Answer: Thank you for these comments, we changed the manuscript accordingly.

Round 2

Reviewer 1 Report

The manuscript has been greatly improved, specifically by the revised introduction and discussion sections. The additional data and the revision of the text in the methods section and the figure legends have greatly contributed to dispelling concerns regarding the conductions of the experiments. One minor point still remains: In Figure 2B the uppermost white arrow of the right panel should be omitted as it indicates a phagolysosome containing a bacterium as can be seen in the image of the left panel (see attached file to illustrate).

Author Response

Dear reviewer,

thank you very much for helping us improve the mansucript with your constructive questions and suggestions. All of them helped a lot! 

Regarding Fig. 2B, we agree and have deleted the arrow and uploaded a new version.

Furthermore, we proofread the manuscript anc corrected some minor spelling mistakes. 

Kind regards

Nicole de Buhr

Reviewer 2 Report

Thanks to the authors for detailed corrections.

Author Response

Dear reviewer,

Thank you very much for helping us with your constructive questions to improve the manuscript.

We have proofread the manuscript and corrected some minor spelling mistakes. 

Kind regards

Nicole de Buhr